# A Review of Sustainable Use of Biogenic Nanoscale Agro-Materials to Enhance Stress Tolerance and Nutritional Value of Plants

**DOI:** 10.3390/plants12040815

**Published:** 2023-02-11

**Authors:** Ved Prakash Giri, Pallavi Shukla, Ashutosh Tripathi, Priya Verma, Navinit Kumar, Shipra Pandey, Christian O. Dimkpa, Aradhana Mishra

**Affiliations:** 1Division of Microbial Technology, CSIR—National Botanical Research Institute, Lucknow 226001, India; 2Department of Botany, Lucknow University, Hasanganj, Lucknow 226007, India; 3Academy of Scientific and Innovative Research (AcSIR), Ghaziabad 201002, India; 4The Connecticut Agricultural Experiment Station, 123 Huntington Street, New Haven, CT 06511, USA

**Keywords:** agro-material, nano-enabled strategies, biological approaches, stress tolerance, plant nutritional value

## Abstract

Climate change is more likely to have a detrimental effect on the world’s productive assets. Several undesirable conditions and practices, including extreme temperature, drought, and uncontrolled use of agrochemicals, result in stresses that strain agriculture. In addition, nutritional inadequacies in food crops are wreaking havoc on human health, especially in rural regions of less developed countries. This could be because plants are unable to absorb the nutrients in conventional fertilizers, or these fertilizers have an inappropriate or unbalanced nutrient composition. Chemical fertilizers have been used for centuries and have considerably increased crop yields. However, they also disrupt soil quality and structure, eventually impacting the entire ecosystem. To address the situation, it is necessary to develop advanced materials that can release nutrients to targeted points in the plant-soil environment or appropriate receptors on the leaf in the case of foliar applications. Recently, nanotechnology-based interventions have been strongly encouraged to meet the world’s growing food demand and to promote food security in an environmentally friendly manner. Biological approaches for the synthesis of nanoscale agro-materials have become a promising area of research, with a wide range of product types such as nanopesticides, nanoinsecticides, nanoherbicides, nanobactericides/fungicides, bio-conjugated nanocomplexes, and nanoemulsions emerging therefrom. These materials are more sustainable and target-oriented than conventional agrochemicals. In this paper, we reviewed the literature on major abiotic and biotic stresses that are detrimental to plant growth and productivity. We comprehensively discussed the different forms of nanoscale agro-materials and provided an overview of biological approaches in nano-enabled strategies that can efficiently alleviate plant biotic and abiotic stresses while potentially enhancing the nutritional values of plants.

## 1. Introduction

Nanotechnology refers to the reshaping of matter at the atomic or molecular level. Nanomaterials are measured in one billionth of a meter (nm) and have a size ranging from 1 to 100 nm [1]. Nanotechnology has evolved as an outcome of the advancement of chemistry, physics, pharmacology, engineering, and biology [2]. Nanomaterials are used in multiple applications, including nanosensors, biosensors, nanochips, biofuels, biolabeling, semiconductors, and antimicrobial agents [3]. Notably, advancements in nanotechnology have provided new scopes in agricultural science and uplifted the agricultural system [4], where the unique properties of nanomaterials make them suitable tools for sustainable agricultural applications [5]. In this regard, agro-materials can be developed into nanoscale materials to aid in fertilizer improvement, either themselves being the active ingredient or used to facilitate the bioactivity of nutrients. Such products are termed nanofertilizers or nano-enabled fertilizers [6]. Usually, fertilizers increase crop yield, whereas crops are characterized by stunted growth and diminished productivity due to improper fertilization and loss of organic matter from the soil [7]. Agriculture is the bedrock for national development in every nation, and there are unique opportunities to develop agro-formulations that can improve the productivity of crops and minimize the risk associated with conducting agriculture under a changing climate [8,9]. To meet the demand of the growing population, a huge amount of agricultural crop production is required. This requirement is currently being fulfilled by the use of conventional chemical-based agrochemicals (chemical fertilizers, pesticides, herbicides, and antimicrobials) [10]. However, excessive use of agrochemicals causes serious problems for both the environment and populations that consume agricultural products or are exposed to them [11], ultimately affecting the food chain [12]. Therefore, there is an urgent need to develop new agro-materials that can overcome the harmful effects of chemical agents and reduce their application in an environmentally safer way. Such nanoscale agro-materials are potentially utilized in different applications, such as nanofertilizers, nanopesticides, nanoinsecticides, nanoherbicides, nanobactericides, bio-conjugated nano complexes, and nanoemulsions, among others. Fortunately, nature functions as a large “bio-laboratory” with an abundance of biomolecule-containing organisms such as plants, algae, fungi, yeast, and others. These naturally occurring biomolecules have been found to actively contribute to the generation of nanoparticles with various surface chemistries, shapes, and sizes, serving as a catalyst for the development of safer, more sanitary, and environmentally friendly protocols for nanomaterials synthesis [13]. The same biological system can be used to create nanoparticles with variable sizes, shapes, and colloidal stability, through different biological activities and changing the synthesis conditions. Microorganisms have a strong enzymatic system that enables them to create nanoparticles of different chemical elements and are capable of manufacturing a large variety of physiologically active substances [14,15]. Despite the many benefits of using nanomaterials in agriculture, there are certain risks associated with large-scale applications. Some of the issues that need in-depth research include ecotoxicity, concentration optimization, impact on soil microbiota, and the safe consumption of exposed plants by animals and humans [16]. However, biogenic nanomaterials can interact with soil components and undergo physical, chemical, and biological transformations that change their stability, reactivity, and toxicity [17]. Some biogenic nanomaterials can be stabilized and persist in soil for a long time, while others can degrade rapidly and become unavailable for environmental interactions. The understanding of the fate and behavior of biogenic nanomaterials in soil systems is still limited, and further research is needed to assess their potential impact on the environment and human health [18]. The cost, scaling-up, and optimization of the biological conditions for the synthesis of nanomaterials are additional steps to ensure the effective use of bioengineered nanomaterials in the field conditions. Strict regulatory guidance regarding the use of nanomaterials should be implemented at a global scale [19]. This review summarizes the recent biological approaches for the synthesis of nanoscale agro-materials that can be applied to efficiently manage the biotic and abiotic stresses in plants, thus potentially enhancing the yield and nutritional value of plant produce.

### 1.1. Applications of Nanoscale Agro-Materials and Their Impact on Plants

Promoting the use of nanotechnology in various facets of the agricultural system is rooted in the fact that the human population is continuously increasing, necessitating the need for sustainable food production. According to population statistics, there will be approximately 9.7 billion people by the end of 2050 (United Nations Department of Economic and Social Affairs, 2015) [20]. Moreover, nutritional deficiencies in food crops increasingly affect human health, particularly in rural areas, due to the inefficiency of plants in utilizing nutrients from conventional fertilizers. Notably, evidence suggests that bio-inspired nanofertilizers that are currently in the research domain are target-oriented and more potent than traditional fertilizers [21]. Chemical fertilizers have been used for centuries and have greatly increased crop yields. However, they cause soil mineral instabilities and destruction of soil structure and quality, as well as manipulate the overall ecosystem, all of which are serious long-term impediments. To overcome the problems, it is necessary to create advanced bioactive materials that can be tuned to release nutrients at specific times [22]. Agronomic biofortification is the action of increasing the nutrient content of food crops with an outcome targeted at the edible plant tissue. Biofortification can be implemented during the nascent stage of plant development [23], and hence, the term agronomic biofortification. According to recent research, nanotechnology has the potential to revolutionize the agricultural system, including by assisting conventional agronomy in fortifying food produce with nutrients [8,24] and facilitating the safe, target-bound delivery of agrochemicals. The properties of nanomaterials, such as the high surface area to volume ratio, may allow for efficient nutrient uptake by crops to maximize yield. The nano-size of nanofertilizers allows them to enter the nano-porous surfaces of the plant tissue, helping to improve fertilizer use efficiency, restore soil fertility, and effectively reduce agroecological degradation. Several nanoscale materials such as zinc oxide (ZnO), copper oxide (CuO), silica (Si), iron (Fe), titanium dioxide (TiO_2)_, zinc sulfide/zinc cadmium selenide (ZnS/ZnCdSe), core-shell quantum dots (QDs), phosphorous/zinc sulphide (P/ZnS) core-shell QDs, manganese-doped zinc selenide (Mn/ZnSe) QDs, gold nanorods, aluminum oxide (Al_2_O_3_), cerium(IV) oxide (CeO_2_), and iron (II) oxide (FeO) are among those used as nanoscale agro-materials [4] (Figure 1).

Nanoscale materials have the potential to influence plant growth, development, and metabolic activity [25,26]. However, it is critical to consider the interaction and impact of nanomaterials in agricultural systems because sometimes they appear with negative effects, dependent on the nanomaterial, dose, and exposure conditions. Among those, Zn and Cu have exhibited toxic effects in a few particular plants, whereas Si has been reported as beneficial to the majority of flora [27]. Similarly, silver nanoparticles (AgNPs) pose both positive and negative impacts on plant growth and development by increasing the peroxidase and catalase activities which catalyze the synthesis of antioxidant complexes. For example, AgNPs improve plant growth and seed germination of *Lolium multiflorum* and *Eruca sativa* [28], while in *Vigna radiata* and *Sorghum bicolor*, they significantly reduce the root length at higher concentrations [29]. Carbon nanotubes (CNTs) possess many unique properties, and they are of two types, namely, single-walled carbon nanotubes (SWCNTs), which act as transporters to transmit DNA fragments, and multiwalled carbon nanotubes (MWCNTs), which improve the nutrient uptake efficiency of the plant [30]. In *Oryza sativa*, *Glycine max*, *Brassica juncea*, *Phaseolus mungo*, and *Lycopersicum esculentum*, MWCNTs were reported to increase the rate of germination and improve the peroxidase and dehydrogenase activities [31,32]. Similarly, in *Glycine max*, nano-silicon dioxide (SiO_2_) and nano titanium dioxide (TiO_2_) increased seed sprouting by increasing the nitrate reductase activity [33]. Haghighi et al. [34] also reported that 1 and 2 mM of nano-silicon increased the seed germination rate and growth of tomatoes by ameliorating the negative effect of salinity. Zinc oxide nanoparticles (ZnONPs) (0–1600 mg/L) efficiently increased the seed germination and root elongation in *Cucumis sativus, Medicago sativa*, *Cicer arietinum*, and *Solanum lycopersicum* [35], while in another study, ZnONPs (500–1500 mg/L) negatively affected the plant growth, antioxidative response, and rate of seed germination in *Brassica nigra* [36]. Collectively, there is ample evidence that several nanomaterials have the potential to increase seed germination, plant growth, and productivity, as well as alleviate biotic and abiotic stresses. Additionally, nanotechnology can enhance nutrient uptake, physiological performance, and minimize metal toxicity in plants, all of which, together, increase agricultural production in an economical and environmentally friendly way [37].

### 1.2. Role of Biogenic Nanoscale Agro-Materials in Nutritional Value-Added Plants

The occurrence of nutrient deficiency in food crops is drastically affecting human health. Several approaches are known that can improve the nutrient quality in food, such as dietary diversification, the use of medicines as a supplement, and industrial fortification during food processing. Due to lower affordability and sustainability, these processes have not yet successfully addressed the problem. Therefore, to fulfill the nutrient requirement, plants require mineral fertilizers that are balanced in both their elemental composition and the amount of each element. On the other hand, the overuse of chemical fertilizers and pesticides is toxic to the environment and leads to serious human and environmental health issues. Agrochemical overuse also exacerbates the high inefficiency of use by plants. The use of nano forms of N, P, K, Fe, Mn, Cu, Mo, and CNTs as fertilizers can improve the bioavailability and targeted delivery of these nutrients in the plant, thereby lending themselves to use under the current scenario. Similarly, nanopesticides such as Ag, Cu, SiO_2_, and ZnO, among others, have better broad-spectrum pest management efficiency as compared to traditional chemical pesticides [21,38].

Presently, considerable efforts by scientists and researchers are underway to advance techniques that could assist plants in enhancing their native functions. Several bioactive compounds (e.g., flavonoids, phenolic acids, alkaloids, and carotenoids) are commercially available as products, with a wide array of applications in the agriculture, food industry, medical, pharmacological, and cosmetic sectors. Researchers are studying the role of bio-inspired nanomaterials as novel elicitors for the biosynthesis of bioactive compounds and potentially promote the plant’s secondary metabolism. Recent studies have noted that the efficient applications of nanotechnology in crop production could enhance the nutritional quality of plants under stress conditions [39]. Particularly, metal oxide nanoscale materials can modulate the plant’s physiological processes to promote the growth of plants. As several of the metals are nutrients, metal oxide nanomaterials can be used as nutrient material and could increase the production of bioactive compounds. The biosynthesized nanomaterials are an appropriate choice due to their biocompatibility, stability, and non-toxic behavior. Some metal oxide NPs, such as titanium oxide, zinc oxide, iron oxide, and copper oxide, have been studied for the development and enhancement of secondary metabolite production in plants [40].

For example, Zn^+2^ is an important micronutrient as well as a co-factor for nutrient-mobilizing enzymes and is essential for crop production and human nutrition [41]. ZnONPs can improve the nutritional quality of the plant and enhance its pharmaceutical value-added constituents. Agronomic biofortification of crop plants is an efficient strategy focused on enhancing and improving the mineral content of staple foods. Karimi et al. [42] synthesized plant-based ZnONPs using *Allium jesdianum* extract; the material possesses anti-cancer activity and can enhance plant growth and physicochemical changes in plant tissue culture. They acted as co-factors for nutrient-mobilizing enzymes and resulted in higher production of secondary metabolites during callus production. Velázquez-Gamboa et al. [43] synthesized ZnONPs using *Moringa oleifera* extract for the biofortification of *Stevia rebaudiana.* In this study, they found that the ZnONPs could enhance the total phenolic (60.5%), flavonoid (87.8%), and zinc contents up to 406.8% in *S. rebaudiana.* Similarly, Del Buono et al. [44] synthesized biogenic ZnONPs using an extract of the aquatic species *Lemna minor* (duckweed). They could enhance the concentration of pigments (anthocyanin), antioxidants (carotenoids), and chlorophylls in maize. Salih et al. [45] synthesized bio-inspired ZnONPs using leaf extracts of *Phoenix dactylifera* L. in particle sizes ranging from 16 to 35 nm. The ZnONPs enhanced the shoot growth and callus development, with improvements in the biochemical contents, including chlorophyll a, total phenolic, and flavonoid contents in *Juniperus procera*. 

Similarly, copper is a critically important microelement for plant growth and development. Cu is available in natural conditions as Cu^2+^ and Cu^+^, and the optimum concentrations required in plants range from 10^−14^ to 10^−16^ M. Cu also plays an important role in cell wall metabolism, protein regulation, and in secondary molecule signaling in plant cells. It is also an obligatory element in mitochondrial respiration, photosynthetic electron transport, ion mobilization, hormone signaling, and oxidative stress response, and it acts as a co-factor for several enzymes [46]. Sarkar et al. [47] biosynthesized copper nanoparticles (CuONPs) using plant extracts of *Adiantum lunulatum*, which can enhance the defense enzymes and the total phenolic content in the roots of *lens culinaris* after seed treatment with 0.025 mg/mL of the CuONPs. Similarly, Jasim et al. [48] synthesized AgNPs using *Bacillus* strain CS 11; subsequently, the NPs enhanced the growth of the Fenugreek plant (*Trigonella foenum-graecum* L.) by improving leaf number, the total yield of the plant, root, and shoot length. Batool et al. [49] phytosynthesized AgNPs (30–100 nm) using aqueous extracts from the leaf of *Euphorbia helioscopia* L. The material was shown to enhance the synthesis of secondary metabolites in sunflower (*Helianthus annuus* L.) plants, wherein a 60 mg/mL concentration of the AgNPs enhanced the biochemical and antioxidant activities, including fatty acid composition (palmitic acid, oleic acid, and linoleic acid), secondary metabolite contents, and defense-responsive enzymatic activities of sunflower plants. Similarly, Chung et al. [50] biosynthesized AgNPs using *Bacillus marsiflavi* (KCCM 41588) that enhanced the production of medicinally beneficial bioactive compounds in bitter gourd. The AgNPs in the cell suspension culture of bitter gourd increased the amount of flavonols (1822.37 µg/g), hydroxybenzoic (1713.40 µg/g), and hydroxycinnamic (1080.10 µg/g) acids. Azadi et al. [51] reported that AgNPs synthesized by using *Dracocephalum moldavica* extract could improve plant growth, carotenoids, and essential oil yield in *Thyme vulgaris* L. during UV-B stress. Shavalibor and Bahabadi [52] biosynthesized AgNPs using *Prosopis farcta* fruit extract, which showed plant growth promotion potential with increased carbohydrate, chlorophyll, phenolic, and secondary metabolite contents in *Melissa officinalis* L. In secondary metabolite contents, a significant amount of rosmarinic acid (approximately 50 mg/g of dry weight) was observed when plants were treated with 60 and 100 ppm of AgNPs. Likewise, Soliman et al. [53] biosynthesized AgNPs by using leaf extracts of blue gum (*Eucalyptus globulus)*, where it was observed that the NPs enhance seed germination and antioxidant enzymes, including catalase, peroxidase, and ascorbate peroxidase as well as glutathione and ascorbate contents in *Zea mays* L., *Trigonella foenum*-*graecum* L., and *Allium cepa* L. Apart from this, several studies have shown that the quality of crops can be improved with the introduction of nanomaterials during crop cultivation because of their easy translocation within the plant system. For instance, Azeez et al. [54] biosynthesized AgNPs using pod extracts of *Cola nitida* with the NPs efficiently enhancing the phytochemical contents (antioxidant, flavonoid, and phenolic compounds) in *Amaranthus caudatus* L. Using 50 ppm of the AgNPs applied during cultivation, these workers observed a significant enhancement (35.80%) in flavonoid contents, in which kaempferol and quercetin were the most abundant. They also found enhanced phenolic contents (68.19%) as well as increased 2, 2-diphenyl-1-picrylhydrazyl (DPPH) antioxidant activity (38.7%).

Away from metals, carbon-based nanoscale materials, namely the fullerene derivative C_60_(OH)_20_, or “fullerol”, has been shown to possess antioxidant, antiviral, and anti-cancerous activities. A water-soluble material, fullerol exposure caused the suppression of accumulation of superoxide and hydroxyl radical-mediated lipid peroxidation as well as the initiation of free radical-scavenging activities [55]. Similarly, Kole et al. [56] treated bitter melon (*Momordica charantia*) with fullerol (C_60_(OH)_20_) NPs and observed improvement in plant biomass (54%), fruit yield (128%), and two anti-cancer phytomedicinal contents, namely, cucurbitacin-B and lycopene, which increased by 74% and 82%, respectively. In addition, the contents of two anti-diabetic phytomedicines, charantin and insulin, increased by 20% and 91%, respectively. 

## 2. Nutritional Value-Added Plants and Their Role in Human Health

### 2.1. Vegetables

Vegetables are the cheapest source of vitamins and amino acids [57]. They contain many essential components, which either cannot be synthesized naturally by the body or their synthesis requires specific factors under certain conditions [58]. Many plant compounds such as flavonoids, sterols, phenols, and glucosinolates play a major role in lowering the disease risk. Vegetables like broccoli, cauliflower, brussels sprouts, turnips, kale, mustard, asparagus, spinach, lettuces, and endives contain phytochemicals that have antioxidant, antifungal, and antiviral activities [59] that play a protective role in countering human diseases such as coronary heart disease, diabetes, high blood pressure, cataracts, degenerative diseases, and obesity [60,61]. Vegetables such as tomatoes, cucurbits, pumpkins, squashes, cucumber, gherkins, onions, shallots, garlic, and chilies contribute to the global food economy and have significant nutritional value. These vegetables and others are consumed in all countries. Therefore, governments need to boost investment in farm production, including providing improved crop varieties and sustainable alternatives to agrochemicals, such as pesticides and fertilizers. Good post-harvest management practices, food safety, and market access have to be facilitated to leverage the economic power of countries [62].

### 2.2. Fruits

Dietary habits are linked to the prevention or otherwise of chronic diseases such as cancer, diabetes, Alzheimer’s, and heart disease [63]. Numerous antioxidants are found in fruits and vegetables that also have been linked to their ability to protect against certain diseases and help neutralize free oxygen radicals. Low levels of antioxidants and vitamins in the blood can increase the risk of cancer mortality [64]. Besides, phytochemicals in fruits, particularly phenolic compounds, are responsible for several health benefits [65]. The amounts of oxidants and antioxidants in humans are kept in balance during normal metabolism, which is critical for maintaining optimal physiological conditions [63]. In certain circumstances, excessive oxidant generation can be harmful, resulting in large-scale oxidative damage to biomolecules (lipids, DNA, and proteins). Notably, fruits consist of phytochemicals that stimulate the immune system, regulate gene expression, and also have antibacterial and antiviral potential [65]. For example, apple fruit contains vitamin C, which provides approximately 4% of total antioxidant activity [60]. Common fruits such as the strawberry, plum, orange, red grape, kiwi fruit, pink grapefruit, white grape, banana, apple, pear, and honeydew melon also possess efficient antioxidant activities [66].

### 2.3. Grain Cereal Staples

Grain cereals are one of the most ancient foods on the planet and are the main component of the human diet. Cereals such as wheat, rice, barley, maize, rye, oats, and triticale, account for the majority of agricultural output. Together, these crops are the most important sources of food for human consumption, providing 50% of dietary protein and energy consumed [67]. In addition to carbohydrates and proteins, other nutrients, such as fat, phospholipids, vitamins, and minerals, are present in cereal grains. Cereals, in general, can reduce cancer and coronary heart diseases [68,69]. Wheat (*Triticum*) is one of the oldest cereal grain crops. The genus contains numerous species, three of which are widely grown worldwide: (*Triticum aestivum* L.), durum (*Triticum durum* Desf.), and spelta (*Triticum spelta* L.). On a global scale, common wheat (*Triticum aestivum* L.) is the most widespread species and the world’s second most extensively produced crop. The fundamental goal of modern wheat farming, especially common wheat (*Triticum aestivum* ssp. *vulgare*), is to produce high-yield cultivars with good baking and nutritional properties [70]. Wheat grain consumption accounts for 19% of all calories consumed by humans worldwide. Wheat is used to make bread, pasta, and other bakery items all over the world. As a result, one of the main goals of cereal farming is to produce varieties with increased protein content [71]. 

Similarly, Triticale (X *Triticosecale Wittmack*) has the potential to play a role in the growth of the healthy food market as well as the development of novel cereal products. Triticale is a cross between the A and B genomes of wheat (*Triticum turgidum* L., *Triticum aestivum* L.) and the R genome of rye (*Secale cereale* L.). However, Triticale has a chemical composition more comparable to wheat than rye; it contains higher levels of most of the nutritional components than wheat. Its protein concentration has long been one of its distinguishing characteristics [72]. 

Barley (*Hordeum vulgare* L.) is one of the oldest cereal crops with an evolutionary link with wheat and rye. Barley plays an important role in human nutrition in the form of flakes and groats [73]. The high soluble dietary fiber content of barley grain has elevated its position as a food ingredient. Regular consumption of barley has been linked to a lower risk of ailments such as chronic heart disease, colon cancer, high blood pressure, and gallstones [74]. The presence of bioactive components of vitamins, minerals, fiber, and other phytochemicals is credited with these medicinal potentials. Furthermore, barley possesses several unique phytochemical features, such as the presence of all eight tocol vitamers, which are rarely found in other cereal grains [70]. The eight tocol vitamers together confer more complete potent antioxidant and neuroprotective, anti-cancer, cholesterol-lowering, and cardioprotective effects. Indeed, the tocotrienols, members of the tocol vitamers, are known to affect numerous pathways associated with tumorigenesis, including the cell cycle, apoptosis, and angiogenesis [75]. 

In addition to the above-mentioned medicinal features, cereals are also a good source of macro and micronutrients, which is important for supplying these nutrients to people in regions of the world where cereals constitute daily staple diets. Such macro and micro minerals are required for optimal human nutrition and are directly reflected in human health. Because the human body is incapable of synthesizing macro and micronutrients, they must be supplied from foods at adequate levels [76]. Notably, cereals contain considerable levels of phosphorus, potassium, calcium, and magnesium, as well as zinc, iron, copper, manganese, molybdenum, and boron [77]. However, the presence of these minerals in the plant is a direct function of the amount and form present in the soil [24,78]. The cell wall of cereal grain also contains β-glucan polysaccharide, which is made of glucose molecules and bonded together to form a lengthy polymer chain. β-glucan helps in managing blood glucose levels, and it also possesses anti-cancer properties [79]. 

Grain legumes such as lentils, soybean, lupine, and beans are also bestowed with high nutritional properties. Lentils contain anti-carcinogenic substances such as lectins, glycosidic saponins, and bioactive peptides and are thus reported to help with reducing the risk of cancer. Lentils contain high levels of polyphenols that confer anti-tumor properties [80]; they are gluten-free and can be used by people suffering from celiac disease. Chickpeas are another legume that has gained popularity in the food system due to their unique capability to promote human health. Chickpeas have recently attracted a lot of attention since they are an excellent source of protein, fiber, carbohydrates, and minerals, all of which contribute to a well-balanced diet. The vegan community especially finds them a great source of protein. They also have a low allergen concentration permitting them to be used as a soy substitute [81]. However, soy is a legume that has a unique place in the vegetarian world with several health benefits [82]. Soy is laden with isoflavones and antioxidative diphenolic chemicals that can help to prevent diseases like osteoporosis, cardiovascular disease, and postmenopausal syndrome [83]. Isoflavones are a type of naturally occurring isoflavonoid which act as phytoestrogenic compounds. Soy protein is a good source of amino acids as well [84]. Lupine, another legume, contains higher (35–40%) protein content than legumes [85]. It has anti-inflammatory properties [86], high lysine content, and is a rich source of proteins and macronutrients such as zinc and iron [87]. Hemoglobin uses iron to deliver oxygen to the body parts; hence it plays a crucial role in the human body [88]. Similarly, zinc is required for cell division, immunological function, wound healing, and glucose metabolism [89]. The consumption of beans has been found in epidemiological studies to lower the risk of cardiovascular disease, diabetes, and cancer, as well as assist in controlling blood sugar levels [90]. These and other graphical depictions of selected nutritionally value-added crop plants and their health benefits are shown in Figure 2.

### 2.4. Ornamentals and Flowering Plants Used as Food Condiments

Some ornamental and flowering plants are used in food preparation because they provide flavor and aroma to food. For example, *Viola odorata* L. is used as a sugar source in syrups [91]. Flowers can be eaten both fresh (e.g., Marigold flower) and in savory dishes with meat and fish. They are also used in drinks (wine and beer), desserts, jellies, and spices [92]. Usually, some plants are known for the flavoring or nutritional potential of their fruits or leaves, while some flowers are also edible and rarely used in cooking, such as passion fruit, chive, and pumpkin [93]. Although the use of edible flowers is still in its infancy, flowers are a natural source of bioactive chemicals [94]. To this end, several species of ornamental plants have been extensively researched, such as *Centaurea cyanus* L. [95], *Chrysanthemum morifolium* Ramat [96], *Hibiscus rosa-sinensis* L. [97], *Lavandula pedunculata* Cav. [94], pansy [93], *Calendula officinalis* L. [98], and *Rosa* spp. [99]. Edible flowers supply antioxidants and essential oils when consumed in their natural form or when minimally processed [100]. The bioactive properties of edible flowers have been associated with the treatment of ulcerative colitis [101], anti-hyperglycemic, anti-cholinergic activity [94], oxidative effects in erythrocytes, and anti-cancerous activity.

## 3. Biogenic Nanoscale Materials as a Nano-Enabled Tool for Stress Alleviation in Plants

Global progress in agricultural crop production may be hampered because of the increased prevalence of environmental stresses [102]. Climatic stresses originate from rising temperatures, heat waves, drought, and the accumulation of heavy metals, while a variety of biotic stresses, such as fungal and bacterial diseases and insect infestation, put pressure on agriculture [103]. These stresses significantly affect crop yields and, for heavy metals, increase the accumulation of different toxic elements in plant tissues, rendering them unfit for consumption by animals and humans, with otherwise serious health problems [104]. Moreover, biotic and abiotic stresses negatively affect the growth and economic expansion of agricultural and horticultural crops [105]. Besides, climatic stress can cause pollen sterility, shriveled seeds, disrupted photosynthetic and respiratory enzyme activities, and an increase in the production of reactive oxygen species (ROS), with negative impacts on plants [106,107]. Notably, biogenic nanomaterials have recently been explored for use in counteracting the damaging effects of a variety of environmental stressors, including heavy metals, drought, salinity, high temperature, and bacterial and fungal pathogens [102]. Green synthesis of nanomaterials gained serious attention in recent years due to their sustainable application in the agricultural system [108]. Several biogenic metallic nanoparticles *viz*; AgNPs, AuNPs, CuNPs, FeNPs, FeS_2_NPs, TiO_2_NPs, ZnNPs, and ZnONPs, have been used to improve seed germination, plant growth, and stress tolerance in a variety of crop plants, as well as in the direct inhibition of plant pathogens, with a view to application in real plant pathosystems [109,110,111]. For instance, biosynthesized AgNPs enhanced plant growth and development by improving seed germination, growth parameters, water content, photosynthetic pigments, osmolytes, and antioxidant pigments [112]. Similarly, Noman et al. [113] reported biogenic CuNPs alleviated salt stress in maize by modulating cellular oxidative repair mechanisms. Del Buono et al. [44] also described biogenic ZnNPs synthesized from *Lemna minor* (duckweed) enhanced the physiochemical and biochemical traits in maize. An overview of nanomaterials-based agri-product and their role in the alleviation of biotic and abiotic stresses is provided in Figure 3. 

Likewise, TiO_2_NPs were shown to mitigate the negative impact of drought stress in wheat, wherein they increased plant height and biomass during drought stress [114]. Similarly, TiO_2_NPs also reportedly increased the starch and gluten content in wheat [115]. In maize seedlings, ZnONPs significantly increased the photosynthesis rate, photosynthetic pigments, transpiration rate, stomatal conductance, and water use efficiency [116] while also increasing uptake of nitrogen in soybean, wheat, and sorghum, even under drought stress where N uptake was inhibited [117,118,119]. Similarly, Dhoke et al. [120] found that iron (II, III) oxide NPs (Fe_3_O_4_NPs) can increase root length and the photosynthetic rate in *Vigna radiata*. As with metal oxide NPs, biogenic nanoemulsions have also gained attention for their application in the agri-sector. Several studies have found a considerable potential in nanoemulsions for plant growth promotion effects attributable to stress amelioration. Priyaadharshini et al. [121] found in pearl millet under drought conditions that a foliar spray of chitosan-based nanoemulsion significantly alleviated drought stress by improving plant water level and yield. Correspondingly, the trichogenic novel lipid nanoemulsion could provide a sustainable approach to managing biotrophic oomycetous pathogens that cause the downy mildew disease in pearl millet [122]. Recently, our group also synthesized a peppermint oil-based nanoemulsion to combat biotic stress in tomatoes during early blight disease caused by the fungal pathogen *Alternaria solani.* We observed that the nanoemulsion formulation could ameliorate the biotic stress at both the physiological and biochemical levels [123]. Relatedly, nanoemulsions from the essential oils *Melaleuca alternifolia* or *Cymbopogon martini* in aloe polysaccharide suspension were used to treat *Xanthomonas fragariae* in strawberries. The treatment significantly reduced the biotic stress caused by the pathogen [124]. Vega-Vásquez et al. [125] synthesized several nanoemulsions by using essential oils, including cinnamon, peppermint, clove bud, oregano, coriander seed, geranium, and red thyme, followed by evaluation and observation of a strong antifungal activity against *Botrytis cinerea*, which is a broad host range necrotrophic phytopathogen that affects different plant species.

### 3.1. Role of Biogenic Nanoscale Materials in Enhancing Stress Tolerance in Plants

#### 3.1.1. Effect under Biotic Stresses

Biotic stresses cause extensive losses in the agri-sector worldwide and raise the risk of hunger in several parts of the globe. Annually, approximately 20–40% of crop yield is lost due to plant diseases caused by several pests and pathogens [126]. Plants are constantly exposed to biotic stresses, including microbial pathogens, insects, and nematodes, which pose economic losses and ecological alterations in cropping systems [127,128]. Biotic stresses can change plant metabolism, including physiological damage, which leads to reduced productivity [129]. Currently, synthetic fungicides and pesticides are used to manage plant diseases or biotic stresses; however, these agrochemicals are harmful to the environment and to human health. Nanotechnology has emerged as an alternative to conventional agrochemical use for the sustainable and eco-friendly management of biotic stresses induced by pests and pathogens on crops [130]. The promise of nanotechnology as an alternative for plant disease and pest management is anchored on the unique properties of nanomaterials: small size, large surface area, tunable solubility, and prolonged residual activity, in addition to other physicochemical properties [131]. Nanotechnology confers several advantages to pesticides by reducing their toxicity and exponential shelf-life with regulated active ingredient availability. Below, we describe reported examples of different biotic stresses and their management using nanotechnology. 

##### Fungal and Bacterial Stress

Fungi and bacteria are primarily responsible for most plant diseases. Fungal parasites may be biotrophs and necrotrophs, which secrete toxins that lead to cellular death. Fungi, together with bacteria, cause vascular wilts, leaf spots, cankers, and several other well-known pathogenic symptoms in different parts of the plant [129]. Kumari et al. [132] biosynthesized AgNPs from the cell-free extract of *Trichoderma viride* (MTCC 5661) and investigated their role against early blight disease caused by *Alternaria solani* in *Solanum lycopersicum*. Mechanistically, the biosynthesized AgNPs could directly kill the pathogens but could also improve the photosynthetic rate and disease resistance while reducing the levels of stress enzyme activities in the tomato plant. Likewise, Ibrahim et al. [133] investigated the impact of biogenic AgNPs (size 25 to 50 nm) on bacterial infection in rice plants and found significant plant growth promotion efficacy of the particles. These NPs were synthesized using the endophytic bacterium *Bacillus siamensis* strain C1 isolated from the medicinal plant *Coriandrum sativum.* Notably, the biogenic AgNPs also could inhibit the growth of other bacteria, such as *Xanthomonas oryzae* pv. *oryzae* (Xoo) LND0005 and *Acidovorax oryzae* (Ao) strain RS-1, the causal agent of bacterial leaf blight and bacterial brown stripe disease in rice (*Oryza sativa* L). The AgNPs remarkably enhanced the root and shoot length and fresh and dry weight of rice seedlings. Kaur et al. [134] studied the role of biogenic AgNPs that were synthesized using the rhizospheric microflora of chickpea (*Cicer arietinum*). The NPs were subsequently applied for wilt disease management in chickpeas caused by *Fusarium oxysporum* f. sp. *ciceri* (FOC). The authors reported the AgNP-coated seeds having a high germination rate of up to 98% as well as a 73.33% reduction in wilt incidence by the AgNPs over the control treatment. No harmful effect of the AgNPs was observed on soil-native microflora at the tested dose. TiO_2_ NPs have also been reported to activate the antioxidant system of the plants [135]. Satti et al. [136] biosynthesized TiO_2_NPs by using aqueous leaf extract from *Moringa oleifera* Lam, followed by evaluation in wheat to combat spot blotch disease caused by the fungus, *Bipolaris sorokiniana*. The measured parameters in the wheat plants significantly increased after treatment with 40 mg/mL foliar applications of TiO_2_NPs. There was an increase in relative water index, membrane stability (29%), protein, and proline contents (10.2 µg/mL). The NPs decreased the contents of soluble sugar as well as phenolic components, ultimately inducing disease resistance in wheat. In addition, the TiO_2_NPs increased the production of carbohydrates, which promoted growth and photosynthetic rate in the disease-stressed plants. Mishra et al. [137] studied the fungicidal efficacy of biosynthesized AgNPs (~12 nm) using *Stenotrophomonas* sp. BHU-S7 (MTCC 5978) for the management of collar rot disease in chickpeas caused by *Sclerotium rolfsii*. The biosynthesized AgNPs protected the plants by reducing the sclerotia germination, inducing the ferulic and myricetin phenolic contents, lignin deposition, and H_2_O_2_ production in chickpeas. Sultana et al. [138] biosynthesized AgNPs (100 nm) using the aqueous extract of *Moringa oleifera* leaves. The authors observed the significant antifungal potential of the AgNPs on rice plants against *Aspergillus flavus*, noting a significant reduction in the synthesis of osmolytes, proline, soluble sugar, total phenol, and flavonoid contents, as well as the activities of superoxide dismutase, peroxidase, and catalase enzymes. 

Hashem et al. [139] synthesized selenium nanoparticles (SeNPs) using *Bacillus megaterium* (ATCC 55000) and investigated the protective effect of the NPs on *Vicia faba* cv, Giza 716 under the biotic stress of *Rhizoctonia solani*. The SeNPs efficiently reduced the damping-off and root rot diseases in *V. faba* and promoted plant growth. Chen et al. [140] phytosynthesized CuONPs using papaya leaf extract and observed that 250 μgmL^−1^ of CuONPs have strong bactericidal activity against *Ralstonia solanacearum*, the causal agent of bacterial wilt in the tobacco plant. The biosynthesized CuONPs significantly decreased biofilm formation, the transcriptional expression of motility-related genes in *R. solanacearum*, and the occurrence of bacterial wilt disease in tobacco plants.

Ponmurugan [141] investigated the antimicrobial activity of biogenic AuNPs against *Phomopsis theae*, the fungus causing canker disease in tea plants. The NPs were synthesized using *Trichoderma atroviridae.* They found that AuNPs considerably reduced the growth of *P. theae* and enhanced the tea leaf yield. Alam et al. [142] biosynthesized Fe_2_O_3_NPs using *Skimmia laureola* leaf extract and observed that 6 mg/mL of the Fe_2_O_3_NPs significantly inhibited the bacterial pathogen *R. solanacearum*, a causal agent of bacterial wilt in plants. The 0.6% *w/v* of Fe_2_O_3_NPs remarkably reduced the bacterial wilt severity in plants.

##### Viral Stress

Viruses can cause systemic damage that leads to stunting, chlorosis, and malformations by affecting the different parts of the plant, while they rarely kill their host [143]. Abdelkhalek and Al-Askar [144] synthesized ZnONPs using *Mentha spicata* aqueous leaf extract. The NPs showed significant antiviral activity against the tobacco mosaic virus, with a foliar application (100 µg/mL) drastically reducing viral accumulation and disease severity up to 90.21%. Treatment with the ZnONPs also altered the transcriptional levels of PAL, PR-1 (salicylic acid marker gene), chalcone synthase, and peroxidase genes, all of which were up-regulated in the plants. Elbeshehy et al. [145] observed a remarkable antiviral activity of AgNPs biosynthesized using *B. pumilus*, *B. persicus*, and *B. licheniformis* against bean yellow mosaic virus infection in *V. faba*, cv. Giza 3 plants. In a related study, the same authors found that the AgNPs could restore the metabolism of infected leaves and also improve plant health by promoting photosynthesis rates and pigment contents [146]. Ogunyemi et al. [147] biologically synthesized the three metal oxide NPs, ZnO, MnO_2_, and MgO, with particle sizes of 62.8, 18.8, and 10.9 nm, respectively, using the rhizospheric bacteria *Paenibacillus polymyxa* strain Sx3. A 16.0 µg/mL concentration of all three NPs significantly reduced the growth and biofilm formation of *P. polymyxa.* Importantly, the application of all three NPs showed significant bacterial leaf blight disease management potential in rice plants. Mishra et al. [148] observed that AgNPs (size ~12 nm) biosynthesized using *Stenotrophomonas* sp. BHU-S7 exhibited strong antibacterial activity against *Xanthomonas oryzae* pv. *oryzae* (Xoo), a causal agent of blight disease in rice. The NPs significantly reduced blight symptoms in rice sheaths. Abdelkhalek et al. [149] found that chitosan/dextran nanoparticles (CDNPs) completely reversed the alfalfa mosaic virus (AMV) symptoms in the *Nocotiana glutinosa* plant. They observed that the foliar application of CDNPs could reduce the disease severity and remarkably decrease the viral accumulation level by up to 70.43% under greenhouse conditions. The NPs also induced systemic acquired resistance and increased total carbohydrates and phenolic contents in the plants. 

##### Parasitic Stress

Nematodes absorb the contents of plant cells and feed on all parts of the plant, causing primarily soil-borne diseases that weaken the plant root system. Nematodes are responsible for nutrient deficiency and the production of symptoms like wilting or stunting [150]. Ghareeb et al. [151] investigated the nematicidal activity of biosynthesized AgNPs against root-knot nematodes *Meloidogyne javanica* in tomato plants. The AgNPs were synthesized using an aqueous extract of *Cladophora glomerata.* They observed that biosynthesized AgNPs had the highest nematicidal activity compared to a commercial nematicide. The NPs significantly reduced gall number, egg masses, females per root system/plant, and mortality in juveniles. Therefore, these biosynthesized AgNPs can be used as a potent nematicide against *M. javanica* as it induces the immune system to defend the plant against nematode infection. El-Ashry et al. [152] biosynthesized silicon nanoparticles (SiNPs) using *Fusarium oxysporum* SM5. The SiNPs were spherical-shaped, with a size of 45 nm, and had a negative surface charge of −25.65 mV. The nematicidal activity of the SiNPs was evaluated on both egg-hatching and juvenile (J2) root-knot nematode *Meloidogyne incognita* in eggplant *Solanum melongena* L. The SiNPs reduced nematode reproduction, egg masses on roots, gall formation, and the final population of J2 in the soil. Similarly, Danish et al. [153] biosynthesized AgNPs using *Senna siamea* and evaluated their efficacy against *Meloidogyne incognita* in *Trachyspermum ammi*. They observed that plants treated had improved growth, with 50 ppm AgNPs showing a strong reduction in gall numbers, egg masses, and root-knot index. Anatomical studies revealed the accumulation of lignin in roots, suggesting an increased root structural barrier against nematode entry.

##### Insect Pests Stress

As with pathogens and nematodes, the quality and quantity of food production are also greatly deteriorated by crop pests such as insects and mites, leading to agricultural losses [154]. Insects and mites damage plants by directly feeding on the plant or by laying eggs within plant tissues. Moreover, piercing or sucking insects act as viral vectors, transmitting viruses into plants through their styles [155]. Green insecticidal NPs have the potential for insect pest management [154]. Kamil et al. [156] evaluated the bioefficacy of AgNPs synthesized using the entomo-pathogenic fungus, *Beauveria bassiana* against the mustard aphid *Lipaphis erysimi* Kait. The NPs showed 60.08% mortality of the aphids. Similarly, Suresh et al. [154] biosynthesized insecticidal AgNPs using *Suaeda maritime* and observed both larvicidal and pupicidal activities against the cotton cutworm *Spodoptera litura*. 

#### 3.1.2. During Abiotic Stresses

Abiotic stress continues to be a serious threat to agriculture, posing consequential agronomical production losses worldwide. The abiotic factors are considered essential components of the natural environment that, however, can be accentuated by anthropogenic events. They are, therefore, intrinsically involved in the real-life determination of crop productivity. Adverse environmental factors such as drought, salinity, extreme temperature, presence of toxic metals, and nutrient deficiency lead to alteration in crop productivity and loss in the availability of cultivable land. Notably, nanotechnology is being considered as a potential tool to combat such deteriorative effects and to develop mitigating measures in response to these adverse environmental factors. 

##### Drought Stress

Drought stress is considered one of the most devastating abiotic factors that alter the soil microbiota and limits the availability of nutrients to plants. Due to the emergence of osmotic stress and the unavailability of essential elements, the heterogeneity of soil is altered under drought stress [157]. Drought-affected plants develop various physiological stress-induced responses, including wilting of leaves, reduction in leaf area, and leaves abscission, which prevents water loss through the transpiration and photosynthesis process [158]. Notably, biogenic nanomaterials have the potential to mitigate drought stress. For e.g., CuNPs (10–100 nm) and AgNPs (≤10 nm) synthesized using onion (*Allium cepa.* L.) extract and tri-sodium citrate exhibited strong plant growth and yield promotion, as well as alleviated the drought stress in wheat. In this study, the plants were grown in a hydroponic system with polyethylene glycol-induced osmotic stress. Application of biogenic CuNPs (3 mg/L) and AgNPs (30 mg/L) increased the chlorophyll content (58.30%), water retention (leaf succulence), leaf potassium content (1.34 kg/mg), along with improved stomatal conductance which, together, is an essential process for the regulation of plant metabolism, including photosynthesis and enzymatic reactions [159].

Similarly, Mustafa et al. [160] biosynthesized TiO_2_NPs using leaf extract of *Moringa oleifera* L. The NPs effectively enhanced plant growth by improving root length (33%), shoot length (53%), fresh weight (48%), and dry weight (44%) under drought stress conditions. The TiO_2_NPs also improved chlorophyll content, relative water content, membrane stability index, antioxidants, and osmolyte contents (proline and sugar). Likewise, Ikram et al. [107] used *Allium sativum* L. as a source of reducing and stabilizing agents for the synthesis of SeNPs. Application of 30 mg/L of the SeNPs in wheat under drought stress significantly increased plant height, shoot length, shoot fresh weight, shoot dry weight, root length, root fresh weight, root dry weight, leaf area, leaf number, and leaf length. 

##### Salt Stress

The presence of high Na^+^ and Cl^−^ ions in the soil creates a low water potential and high osmotic pressure in the plant cell, leading ultimately to nutrient and water deficit conditions in the plant. Moreover, high levels of Na^+^ ions impede potassium and calcium homeostasis in plants, along with altered stomatal regulation resulting in reduced photosynthetic efficiency [161]. The prospection and application of biogenic nanomaterials in agriculture to combat salt stress have gained attention due to their eco-friendly, cost-effective, and sustainable approach. Biogenic copper nanoparticles CuNPs (22.44–44.26 nm) synthesized from *Klebsiella pneumoniae* were applied (100 mg/kg) on maize under a saline-treated potted condition, and there was an increase in plant morphological parameters, namely, root length (43.52%), shoot length (44.06%), fresh weight (46.05%), and dry weight (51.82%). Improvement in the antioxidant activity along with the reduction in ionic contents, i.e., Na^+^ and Cl^−^, was observed, indicating the role of nanomaterial in the alleviation of abiotic stress [113]. AgNPs of spherical and semispherical shapes and with 15–30 nm size distribution were synthesized by employing *Capparis spinosa* as the source of reducing and stabilizing agents. The NPs showed the potential to reduce the effect of salinity-induced stress in *T. aestivum* L. Priming of AgNPs on *T. aestivum* seeds improved seed germination (90%), along with increasing the root and shoot length (22% and 58%, respectively), photosynthetic efficiency (42%) and maintenance of the hormonal equilibrium in the plant [162].

Similarly, Mustafa et al. [163] observed the impact of biosynthesized TiO_2_NPs on the morphological and physiological parameters of wheat during salinity stress. The authors found that the application of 40 mg/L of the biosynthesized TiO_2_NPs remarkably improved the physical and physiological parameters, including plant height, dry and fresh weight, shoot and root length, number of leaves, relative water content, membrane stability index, chlorophyll a and b, and total chlorophyll contents, under salinity stress. Likewise, El-Gazzar et al. [164] biosynthesized TiO_2_NPs using *Aspergillus flavus* KF946095 and found that the particles promoted salinity stress tolerance in *Phaseolus vulgaris* L. up to 110%, compared to the control, when tested at a salinity level of 100 mM NaCl. The authors also observed the molecular intensity ratio and the relative density of TiO_2_NPs inoculated arbuscular mycorrhiza fungi (AMF) was higher than AMF alone.

##### Heavy Metals Stress

Advances in human anthropogenic activities in various industries, such as mining and agriculture, have heightened the harmful effects of industrialization on the environment. Such activities increase the content of heavy metals and other toxic elements contamination in various natural environments like soil, water, and air [165]. Heavy metal contamination poses potential detrimental effects and imposes toxicity in animals and plants. Heavy metals and metalloids such as arsenic (As), lead (Pb), cadmium (Cd), mercury (Hg), and Cu are naturally occurring hazardous metals that could accumulate in plant tissue, leading to toxicity that manifests in the inhibition of growth and yield, and the contamination of edible plant portions such as leaves, grains, and fruits. Accumulation of these elements in the edible plant parts compromises the food chain [166]. Notably, green synthesis of nanomaterials is promising for the bioremediation of hazardous metals in the environment, as shown in several studies. For example, heavy metal adsorption via biosynthesized magnesium oxide nanoparticles (MgONPs) was assessed by Fouda et al. [167]. These authors noted that MgONPs (nanorods 30–85 nm, nano-rectangular 18.6–27.6 nm) synthesized using the secondary metabolites of *Aspergillus niger* could remove/adsorb the toxic heavy metal ions of Cr (94.2%), Co (63.4%), Pb (72.7%), Cd (74.1%), and Ni (70.8%) from textile and tannery effluents (wastewater). Further, the toxicity of treated effluent has been evaluated by germinating corn (*Zea mays* L.) and broad bean (*V. faba*) seeds. Significantly decreased phytotoxicity levels were observed in the treated wastewater samples, as compared to the untreated control water sample. Khan et al. (2021b) [168] observed that the application of 5 ppm *Bacillus subtilis*-synthesized Fe_3_O_4_NPs (size 67.28 nm and spherical shape) on rice plants under arsenic toxicity significantly alleviated arsenic stress and improved plant growth. 

El-Saadony et al. [169] biologically synthesized SiNPs by using *Aspergillus tubingensis* AM11 and observed their effect on *Phaseolus vulgaris* L. grown on saline soil contaminated with heavy metals. They found that the bio-SiNPs significantly promoted plant growth and yield, chlorophyll content, carotenoid content, transpiration rate, net photosynthetic rate, membrane stability index, relative water content, stomatal conductance, free proline, total soluble sugars, and defense enzyme responses. In addition, the bio-SiNPs remarkably reduced the electrolyte leakage, malondialdehyde, H_2_O_2_, O_2_^•−^, Na^+^, Pb, Cd, and Ni levels in leaves and pods of *P. vulgaris* L. A 5 mmol/L concentration of the bio-SiNPs was more potent in improving plant performance and decreasing heavy metals content in plants grown in heavy metal-contaminated saline soil. Ragab and Saad-Allah [170] bio-fabricated sulfur nanoparticles (SNPs) using *Ocimum basilicum* leaf extract and investigated their effect on Mn-stressed *Helianthus annuus* (L.) seedlings. Upon treatment of the plants with the SNPs, the authors recorded a significant reduction in Mn toxicity and uptake, with increased sulphur metabolism. The NPs enhanced cysteine levels and improved the seedling’s growth, photosynthetic pigments, and mineral nutrients content.

##### Extreme Temperature Stress

Temperature plays a vital role in the growth and regulation of plants, and the plasma membrane is the plant part most affected by temperature extremes (low or high). As the temperature declines, the lipid content of plants is majorly affected due to low-temperature stress. The water molecules increase as the temperature goes down, resulting in decreased water potential and creating water deficit conditions in the plant cell. The tolerance to low-temperature stress depends upon unsaturated fatty acid content and the downregulation of aquaporins present in the plasma membrane [171]. In contrast, the high temperature tends to affect plant metabolic homeostasis and photosynthetic efficacy and causes severe damage to crops that affect productivity [172]. Although very few studies are available on the role of biogenic NPs in the mitigation of heat stress, in general, biogenic nanoparticles may alleviate heat stress through antioxidant regulation and thylakoid membrane homeostasis. A study performed by Djanaguiraman et al. [173] revealed the role of chemically synthesized selenium nanoparticles (<100 nm), which have the potential to mitigate heat stress in sorghum plants. Foliar application of the NPs enhanced the antioxidant enzyme content, including those of SOD (22%), CAT (24), POX (11), and GPX (9%), while causing a decline in the free radicals O_2_ (29%), H_2_O_2_ (38%), MDA (39%), and membrane damage (25%), compared to untreated plants.

The foregoing explores the potential of biogenic nanomaterials to ameliorate the biotic and abiotic stress in plants; studies on these aspects are listed below in Table 1.

### 3.2. Mechanistic Overview of Different Biogenic Nanoscale Materials during Biotic and Abiotic Stresses

Climate change is more likely to have a negative impact on productive assets and crop yields in developing countries around the world [102], and it affects the production of staple crops such as rice, wheat, and barley [106,107]. The major climatic stresses, including high temperature, drought, and accumulation of heavy metals, and the biotic stresses, namely fungal and bacterial disease infections, significantly reduce agriculture productivity [103]. To combat these problems, some technologies like plant breeding and genetic engineering are being implemented, with certain limitations. Notably, nanotechnology-based interventions are being strongly encouraged in order to complement more established technologies in the effort to meet the world’s growing food demand [188,189]. Nanoscale materials have been administered via foliar spray or by incubating seeds, pollen, and protoplasts, while other methods used to make the NPs available to plants include direct injection, hydroponic treatment, and biolistic delivery [190]. Nanomaterials can penetrate the plants through the root epidermis or by aerial surface via both apoplastic and symplastic pathways [191,192]. If nanomaterials are supplied on the plant’s aerial part, they enter via stomata, wounds, trichomes, stigma, hydathodes, cuticles, or lenticels and are subsequently transported by the phloem. Their entry is also favored by carrier proteins such as aquaporins and other transporters [190]. In particular, the uptake of carbon and metal-based nanomaterials (NMs) is a very nascent area of study. Carbon-based NMs such as fullerene C70, fullerol (C_60_(OH)_20_), and carbon nanotubes are the most studied materials, whereas the metal-based NMs, TiO_2_, Au, Ag, Cu, CeO_2_, FeO, and ZnO NPs are attracting the attention of researchers. Uptake, translocation, accumulation, and toxicity of nanoscale materials depend on the plant species and size, type, chemical composition, functionalization, and stability of the NMs. Particles typically enter the plant root system via the lateral root junctions and reach the xylem through the cortex and pericycle [193]. After internalization, NMs interact with sulfhydryl and carboxyl groups and alter the protein activity. The NMs may form complexes with membrane transporters or root exudates before being transported into plants [194,195] and may move from the leaves to the roots, stem, and developing seeds. The xylem is one of the primary pathways for plant uptake and transportation to the shoot and leaves [196,197]. The NMs can also pass through the leaf cuticle and cell cytoplasm [198] and through the plasmodesmata [199].

Under abiotic stress conditions, plants exhibit an ability to deal with adverse environmental conditions such as chilling, salinity, drought, and heat [200,201]. The response of plants to abiotic stress can be varied and includes increased cytoplasmic Ca^++^, secondary messengers, ROS, abscisic acid, and mitogen-activated protein kinase (MAPK) pathways [202]. These stress responses can also be concomitant with the regulation of proteins involved in cellular damage and defense-responsive genes [203]. Secondary metabolites also help plants to combat abiotic stresses by steadying the cellular structures, signal transduction, polyamine biosynthesis, and protecting photosystems from ROS [204]. ROS commonly act as stress signals and activate the plant’s defense system, as well as enable the plant to protect itself from cellular damage. Notably, NMs also induce the production of antioxidant enzymes, namely, SOD, CAT, and POD, to scavenge ROS. 

Therefore, the application of NMs in the food-agriculture-environment nexus, which exploits the peculiar properties of NMs, can protect and improve the plant’s quantitative and qualitative indices, allowing it to achieve nano-enabled sustainable agriculture. Evidently, nanoscale agro-materials can mitigate both biotic and abiotic stresses and have significant potential to develop stress tolerance in plants, thereby enhancing agricultural production and promoting food security even in the face of climate change. Such protective roles for the different nanoscale materials in improving the plant’s behavior during biotic and abiotic stress are exemplified in Figure 4.

## 4. Conclusions

It is abundantly evident that lowering the application rate and frequency of environmentally harmful agrochemicals currently used for plant protection and growth promotion requires the agricultural system to adopt greener strategies to achieve sustainable agriculture with minimal agrochemical inputs. The application of biogenic nanomaterials has shown strong potential as an alternative to conventional fertilizers and pesticides due to their enhanced compatibility with the environment with high product throughput. However, the real-life commercial application of nanomaterials is still far-fetched. Exposure dose and exposure to environmental conditions have been shown to play a critical role in plant response to NPs and together could determine when an NP is toxic or beneficial [197]. Based on this scientific evidence, guidelines are required from the regulatory bodies specifying the proper methods for safely packaging and applying (dose, exposure route) nanotechnology-based agrochemicals in the field and greenhouse growth systems. As recent research findings continue to demonstrate the use of green nanomaterials and their possible mechanism in the plant during environmental stresses and in soil suffering from nutrient-deficient conditions, more rigorous research and awareness program are critically required to gain the trust of the farmers and produce consumers. A significant portion of the world’s arable soil suffers from nutrient depletion. The consumption of nutrient-deficient food crops significantly affects human health; current strategies to mitigate the effect, such as breeding and genetic engineering, have yet to successfully achieve the desired outcome due to issues of affordability and sustainability. Notably, nanotechnology has emerged as a potential tool to combat such deteriorative effects by its promise of facilitating mechanisms that enhance plant response to adverse environmental conditions. In particular, the use of biogenic nanomaterials is rapidly emerging as an alternative and eco-friendly approach for the sustainable protection of crops, thereby improving both the yield and the nutritional value of plant produce, thus simultaneously promoting food and nutrition security. 

## Figures and Tables

**Figure 1 plants-12-00815-f001:**
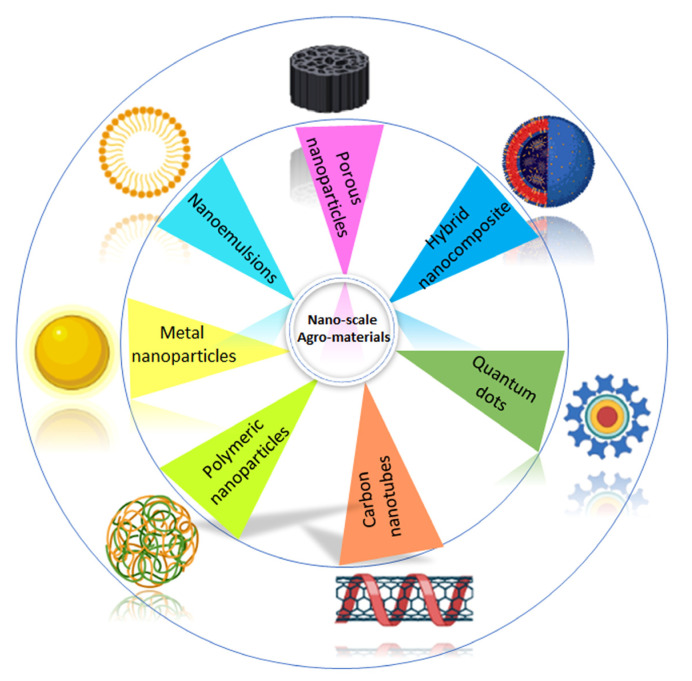
Schematics of different types of nanoscale agro-materials used in agriculture. Image partially created with BioRender.com.

**Figure 2 plants-12-00815-f002:**
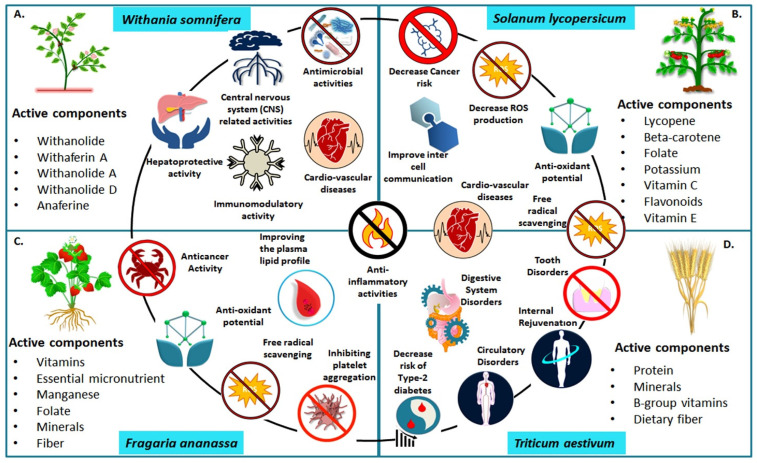
Overview of the nutritional value-addition and health benefits of selected crop plants *viz*; (**A**) Ashwagandha (*Withania Somnifera*) (**B**) Tomato (*Solanum lycopersicum*) (**C**) Strawberry (*Fragaria ananassa*) (**D**) Wheat (*Triticum aestivum*).

**Figure 3 plants-12-00815-f003:**
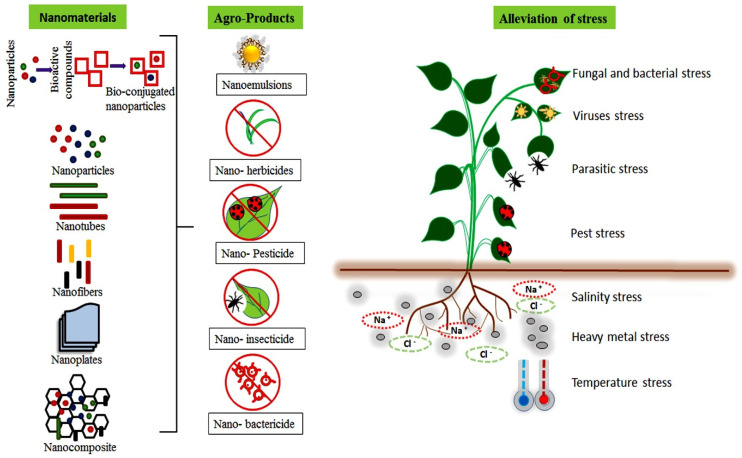
Different bioactive nanoscale agri-products for the alleviation of biotic and abiotic stresses in plants.

**Figure 4 plants-12-00815-f004:**
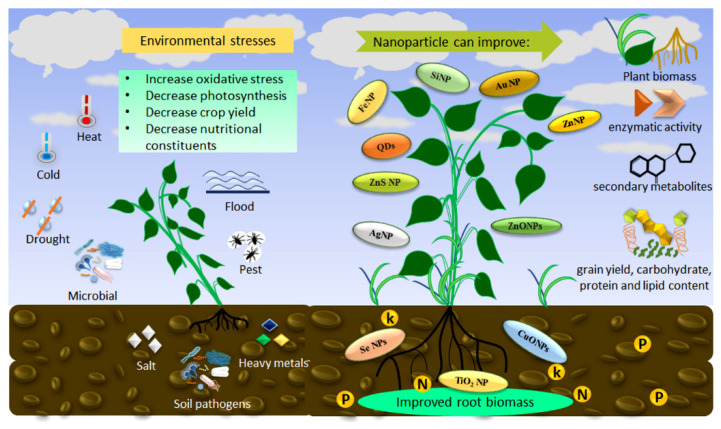
Overview of different environmental stresses and role of nanoscale materials to improve plant response.

**Table 1 plants-12-00815-t001:** Biogenic nanoscale agro-materials used to enhance the stress tolerance and protective management of plant disease.

Sr. no.	Biogenic Nanomaterials	Host Plant	Type of Stress	Function of Nanoscale Materials	References
1.	Copper nanoparticles from *Klebsiella pneumoniae*	*Zea mays* L.	Salt stress	Modulating the cellular oxidative repair mechanism.	[174]
2.	Iron oxide nanoparticles from *Pantoea ananatis*	*Triticum aestivum*	Cadmium and salinity stress	Biomass, antioxidant concentrations, and photosynthetic pigments were increased.	[175]
3.	Peppermint oil nanoemulsion	*Solanum lycopersicum*	Biotic stress by *Alternaria solani* causes early blight disease	Inhibit the spore count and trigger first line of defense.	[123]
4.	Magnesium oxide nanoparticles from *Enterobacter* sp. RTN2	*Oryza sativa*	Arsenic stress	Reduce arsenic tolerance and increase plant biomass and antioxidant activity.	[176]
5.	Selenium nanoparticles from *Citricoccus* sp.	*Chlorella vulgaris*	UV-C irradiation stress	Increases antioxidant defense system.	[177]
6.	Silver nanoparticles from *Trichoderma viride*	*Arabidopsis thaliana*	Biotic stress, black spot disease caused by *Alternaria brassicicola*	Eliciting immunity by altering plant defense proteome and metabolome.	[178]
7.	Silver nanoparticles from *Senna siamea*	*Trachyspermum ammi* (L.)	Biotic stress by nematode *Meloidogyne incognita*	Increases the plant growth and defense system.	[153]
8.	Silver nanoparticles from *Moringa oleifera*	*Oryza sativa*	Biotic stress by *Aspergillus flavus*	Significantly increases the protein content.	[138]
9.	Silica nanoparticles	Eggplant	Biotic stress by nematode *Meloidogyne incognita*	Nematicidal activity and plant growth activity.	[152]
10.	Silver nanoparticles from *Capparis spinosa*	*Triticum aestivum*	Salt stress	Increases plant tolerance by balancing plant hormones and physiological parameters.	[162]
11.	Iron oxide nanoparticle from *Bacillus* strain RNT1	*Oryza sativa*	Drought and cadmium stress	Acropetal Cd translocation and increased nutrient uptake.	[175]
12.	Copper nanoparticles from *Klebsiella pneumonia.*	*Triticum aestivum*	Cadmium stress	Cr translocation to aerial part and increased plant growth.	[174]
13.	Magnetite nanoparticles from *Hevea.*	*Oryza sativa*	Metal stress	Absorption of metal and immobilize it into soil.	[179]
14.	Silver nanoparticles from *Trichoderma viride*	*Solanum lycopersicum*	Biotic stress, early blight disease caused by *Alternaria solani*	Decrease the level of biotic stress revealed by the reduction of enzymatic responses and spore count.	[132]
15.	Selenium nanoparticles from *Bacillus* sp. MSh-1	*Brassica napus*	Cadmium stress	Scavenging ROS production and decreasing Cd accumulation with maintaining calcium homeostasis.	[180]
16.	Silver nanoparticles from rice extract	*Oryza sativa*	Biotic stress, sheath blight disease caused by *Rhizoctonia solani*	Reduce the disease incidence by fungal growth inhibition and improve seedling vigor index.	[181]
17.	Thymol nanoemulsion	*Glycine max*	Biotic stress, bacterial pustule disease caused by *Xanthomonas axonopodis* pv. *glycine*	Plant growth-promoting activity with inhibition of pathogen growth.	[182]
18.	Iron oxide nanoparticles/magnetite from *Chaetomorpha antennina*	*Setaria italica*	Drought stress	Producing photo-assimilates and increases the chlorophyll and sugar content.	[183]
19.	Titanium oxide nanoparticles from *Moringa oleifera* Lam. and calcium phosphate	*Triticum aestivum*	Drought stress	Enhanced nutrient uptake and maintains hormonal level.	[160]
20.	Selenium nanoparticles by using extract of *Allium sativum* L.	‘Kinnow’ mandarin plant	Yellow dragon disease	Improve plant physiology and increases the enzymatic and non-enzymatic antioxidant molecule.	[184]
21.	Zinc oxide nanoparticles from *Halimeda tuna*	*Gossypium hirsutum* L.	Phosphorus utilization	Interact with meristematic cells, trigger biochemical pathways and accumulates biomass.	[185]
22.	Silver nanoparticles from *Phyllanthus emblica* L.	*Triticum aestivum* L.	Ozone-induced stress	Enhance tolerance by increasing biochemical and physiological responses.	[186]
23.	Copper nanoparticles from *Klebsiella pneumoniae* strain NST2	*Zea mays* L.	Salt stress	Increase tolerance by activating antioxidative machinery.	[113]
24.	Iron oxide nanoparticle from *Bacillus* strain RNT1	*Oryza sativa*	Drought and cadmium stress	Acropetal Cd translocation and increased nutrient uptake.	[187]

## Data Availability

All articles reviewed are available online.

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
