# Peer review of "A Review of Sustainable Use of Biogenic Nanoscale Agro-Materials to Enhance Stress Tolerance and Nutritional Value of Plants"

_plants, 2023, doi:10.3390/plants12040815_

Round 1

Reviewer 1 Report

Many thanks to the authors for their effort on this manuscript, but I have few notes;

-It would have been better to have a separate title to talk about the biological synthesis of nanomaterials and what distinguishes it from other physical and chemical synthesis methods.

-In part number 2. Nutritional value-added plants and their role in human health:

I expected the author to link the role of nanoscale agro-materials in improving the nutritional value of crops. It would be better to shorten this part and rephrase it by mentioning agricultural nano-materials and their role in improving the nutritional value of crops.

Typo errors:

Line 44: “antimicrobials” correct to “antimicrobial”

Line 65, 66: delete the duplicated word “nano-herbicides

Line 110: correct “multifolium” to “ multiflorum”

Line 123: correct “sativa” to “sativus”

Line 123: correct” arientum” to “ arietinum”

Line 127: correct “germinations” to “germination”

Line 129: “nutrient depletion”, Rephrase correctly

Line 190: “mgmL”, insert space “mg mL”Line 195: correct “mg/ml” to “mg/mL”

Line 212: correct “globules” to “globulus”

Line 346: correct “Aswagandha” to “Ashwagandha”

Line 385: “improvingregulating”, insert space

Line 388: correc” Buono et al” to “Del Buono et al”

Line 416: correct ” altenifolia” to “ alternifolia”

Line 461: correct “arientium” to “arietinum”

Line 482: write “noting” in normal  style, not Italic

Line 587: “03”. Correct number

Table 1. no 12, correct “Cr” to” Cd”

Lines 120-122:add reference number (26)

Lines 276, 277: add numbering for references

Line 177-180: write the plants whose properties improved after being treated with zinc oxide nanoparticles

References:

Lines 73-75: add reference

The reference list;

Check the references and write the scientific names of plants and microorganisms in italics

Reference No. 64, All letters of the authors' names are capitalized, correct

The attached file may be useful

Author Response

Comments and Suggestions for Authors:

Many thanks to the authors for their effort on this manuscript, but I have few notes;

-It would have been better to have a separate title to talk about the biological synthesis of nanomaterials and what distinguishes it from other physical and chemical synthesis methods.

Response: I appreciate your excellent proposal. The title has been reframed through the inclusion of your perspective.

-In part number 2. Nutritional value-added plants and their role in human health:

I expected the author to link the role of nanoscale agro-materials in improving the nutritional value of crops. It would be better to shorten this part and rephrase it by mentioning agricultural nano-materials and their role in improving the nutritional value of crops.

Response: Thank you for your valuable comments. In section 2, we have comprehensively described the nutritional value of plants only which are generally used in routine diet of human. These plants require mineral fertilizers that are balanced in both their elemental composition and in the amount of each element. The consequences of nutritional deficiencies in food crops are grave for human health. Biogenic nanoscale agro-material a sustainable approach is known that can improve the nutrient quality in food. Though, the role of nanoscale agro-materials in improving the nutritional value of crops also has been discussed extensively in previous section 1.2.

Typo errors:

Line 44: “antimicrobials” correct to “antimicrobial”

Line 65, 66: delete the duplicated word “nano-herbicides”

Line 110: correct “multifolium” to “ multiflorum”

Line 123: correct “sativa” to “sativus”

Line 123: correct” arientum” to “ arietinum”

Line 127: correct “germinations” to “germination”

Line 129: “nutrient depletion”, Rephrase correctly

Line 190: “mgmL”, insert space “mg mL”Line 195: correct “mg/ml” to “mg/mL”

Line 212: correct “globules” to “globulus”

Line 346: correct “Aswagandha” to “Ashwagandha”

Line 385: “improvingregulating”, insert space

Line 388: correc” Buono et al” to “Del Buono et al”

Line 416: correct ” altenifolia” to “ alternifolia”

Line 461: correct “arientium” to “arietinum”

Line 482: write “noting” in normal  style, not Italic

Line 587: “03”. Correct number

Table 1. no 12, correct “Cr” to” Cd”

Lines 120-122: add reference number (26)

Lines 276, 277: add numbering for references

Line 177-180: write the plants whose properties improved after being treated with zinc oxide nanoparticles

Response: Thank you for very keen observation and point out the typo errors. We have corrected the all mistakes in revised manuscript.

References:

Lines 73-75: add reference

Response: Reference has been cited in revised manuscript.

The reference list;

Check the references and write the scientific names of plants and microorganisms in italics

Reference No. 64, All letters of the authors' names are capitalized, correct.

Response: All references have been cited by the Mendeley Reference Manager using the Plants MDPI format. Though, the final formatting will be done in the proofread file because when we add any new references, all references are instantly converted to their own style.

Reviewer 2 Report

I congratulate the authors and suggest further research in the context of the proposals made, in particular on the use of nanofertilizers or nano-enabled fertilizers  in organic farming and their effects on crops.

Author Response

Comments and Suggestions for Authors: I congratulate the authors and suggest further research in the context of the proposals made, in particular on the use of nanofertilizers or nano-enabled fertilizers in organic farming and their effects on crops.

Response: We are very thankful to the reviewer for encouraging remarks and great suggestion.

Reviewer 3 Report

Biogenic Nanoscale Agro-Materials is very interesting and publication summarizing the current state of knowledge is strongly needed.

 The manuscript can be of interest to wide readers of journals and contributes to existing knowledge on the subject matter. However, I have pointed out few pertinent points for improving the clarity of the content and boosting the scientific soundness of the manuscript.

Abstract

Abstract section should described in parts such as- background, methods, results and conclusions.

Introduction not complete. More information may be added on pertinence of following points

l  Occurrence and diversity of biogenic nanoparticles,

l  synthesis of biogenic nanoparticles, and their fate in soil ecosystem,

• general idea on potential applications of nanoparticles in the agricultural systems,

• Impact of biogenic nanoparticles on biotic stress to plants, antimicrobial effects and cytotoxicity,  plant disease management,

• Importance of nanotechnology agriculture with respect to soil, water and plant sciences

Materials and methods

Author must include the description of method  of preparation literature survey in manuscript. Such description should justify why the publications were included into review. Which databases were used (if any)? Which time period was taken into account? The procedure of exclusion/inclusion of literature sources should be described.

 Minor revisions

                  Line 385: “improvingregulating”

                  Line 461, 651,  the scientific name format

(cicer arientium) zea mays

                  Line 584, 592, 597, 618, 630, 658, 668: Scientific name should be in italic form. But, author (name(s) of the person or persons who first published that name for the plant) should not be in italic form. Check it throughout the manuscript. There is no consistency.

Table 1: Number 19 and 20: Titanium oxide nanoparti-

cles from” and “Selenium nanoparticles by using extract of” should not be in italic form

Author Response

Reviewer 3

Biogenic Nanoscale Agro-Materials is very interesting and publication summarizing the current state of knowledge is strongly needed. The manuscript can be of interest to wide readers of journals and contributes to existing knowledge on the subject matter. However, I have pointed out few pertinent points for improving the clarity of the content and boosting the scientific soundness of the manuscript.

Response: Thank you for the appreciation and encouraging remarks on the manuscript.

Abstract: Abstract section should described in parts such as- background, methods, results and conclusions.

Response: I appreciate your feedback, but the abstract portion is formatted in accordance with previously published review article and journal guidelines.

Introduction not complete. More information may be added on pertinence of following points

l Occurrence and diversity of biogenic nanoparticles, l  synthesis of biogenic nanoparticles, and their fate in soil ecosystem.

  • general idea on potential applications of nanoparticles in the agricultural systems,
  • Impact of biogenic nanoparticles on biotic stress to plants, antimicrobial effects and cytotoxicity, plant disease management,
  • Importance of nanotechnology agriculture with respect to soil, water and plant sciences

Response: The introduction section has been revised through the inclusion of some necessary information in order to your useful suggestions.

Materials and methods:

Author must include the description of method of preparation literature survey in manuscript. Such description should justify why the publications were included into review. Which databases were used (if any)? Which time period was taken into account? The procedure of exclusion/inclusion of literature sources should be described.

Response: Thank for the suggestions. In this review we did not used any particular database and neither confined any specific time period. We conducted a careful evaluation of the literature on the main biotic and abiotic stressors that affect plant growth and production. We have thoroughly discussed the various forms of nanoscale agro-materials, such as nanopesticides, nanoinsecticides, nanoweedicides, nanobactericides, bio-conjugated nano complexes, nanoemulsions, and nanoherbicides in terms of target orientation and stress management in plants over conventional fertilizers and chemical pesticides, because they hold immense potential for achieving agricultural sustainability. This review is essential to ensure that future research is focused on addressing the creation of nanoscale agro-materials and assessing their effectiveness in enhancing crop productivity and nutritional value.

Minor revisions

-Line 385: “improvingregulating”

-Line 461, 651, the scientific name format “(cicer arientium)” zea mays

-Line 584, 592, 597, 618, 630, 658, 668: Scientific name should be in italic form. But, author (name(s) of the person or persons who first published that name for the plant) should not be in italic form. Check it throughout the manuscript. There is no consistency.

-Table 1: Number 19 and 20: Titanium oxide nanoparticles from” and “Selenium nanoparticles by using extract of” should not be in italic form.

Response: Thank you for the keen observation and very important comments. All the suggestions have been incorporated in the revised manuscript.

Finally, authors are grateful to the reviewers for their critical examination of manuscript for the sole purpose of empowering it. We are highly thankful to them and to the entire Plants MDPI editorial board.

Round 2

Reviewer 3 Report

The author has revised the manuscript carefully and responded to all the comments. Now the manuscript can be accepted in its present form.